# Mechanisms Controlling the Expression and Secretion of BDNF

**DOI:** 10.3390/biom13050789

**Published:** 2023-05-02

**Authors:** Juan Carlos Arévalo, Rubén Deogracias

**Affiliations:** 1Department of Cell Biology and Pathology, Institute of Neurosciences of Castille and Leon (INCyL), University of Salamanca, 37007 Salamanca, Spain; 2Institute of Biomedical Research of Salamanca (IBSAL), 37007 Salamanca, Spain

**Keywords:** BDNF, proBDNF, Ca^2+^, expression, nerve cells, neurotrophins, neurotrophin release, secretion, synaptic activity

## Abstract

Brain-derived nerve factor (BDNF), through TrkB receptor activation, is an important modulator for many different physiological and pathological functions in the nervous system. Among them, BDNF plays a crucial role in the development and correct maintenance of brain circuits and synaptic plasticity as well as in neurodegenerative diseases. The proper functioning of the central nervous system depends on the available BDNF concentrations, which are tightly regulated at transcriptional and translational levels but also by its regulated secretion. In this review we summarize the new advances regarding the molecular players involved in BDNF release. In addition, we will address how changes of their levels or function in these proteins have a great impact in those functions modulated by BDNF under physiological and pathological conditions.

## 1. Introduction

Since its purification and characterization from the pig brain by Barde and collaborators, brain-derived neurotrophic factor (BDNF) has been related with many different functions in both the central (CNS) and the peripheral (PNS) nervous system, as well as with CNS pathologies, for what is viewed as an endogenous “neuroprotectant” [1,2,3,4].

As all four members of the neurotrophin family, which also includes nerve growth factor (NGF), neurotrophin-3 (NT-3), and neurotrophin-4 (NT-4), BDNF is synthesized and folded in the endoplasmic reticulum as a glycosylated precursor, the pre-proBDNF. This is a highly conserved protein that contains a small signal peptide, a pro-domain, and a mature domain responsible for most of the biological effects associated with BDNF. Once the proBDNF has been synthesized at the endoplasmic reticulum, it is transported to the Golgi apparatus. Under non-pathological conditions, this isoform undergoes an endoproteolytic cleavage [5,6], which promotes its targeting to secretory granules, namely dense core vesicles (DCVs) [7]. The proteolytic cleavage of proBDNF seems to predominantly be performed by intracellular proteases such as furin, and prohormone proconvertases PC1, PC2, and PC7, but can also occur once it is secreted by extracellular proteases such as plasmin and by matrix-metalloproteinases [8,9,10,11,12,13,14,15]. BDNF and its pro-domain bind each other with high affinity in the trans-Golgi network (TGN), suggesting that both move together [16]. It has also been reported that the Golgi-resident targeting receptor sortilin binds to the BDNF pro-domain, sorting BDNF to the regulated secretion pathway [17]. It has been reported that the vesicle membrane-resident sorting receptor carboxypeptidase E (CPE) controls transportation of BDNF into secretory granules [18]. 

Contrary to other growth factors, which are primarily secreted via a constitutive pathway, most of the newly synthesized BDNF is sorted into a regulated pathway that depends on neuronal activity and Ca^2+^-dependent signaling [19,20,21]. In addition, the sequence information located in the BDNF pro-peptide sequence as the one located in the mature portion participates in BDNF targeting [18,22]. Interestingly, the BDNF Val66Met polymorphism, found in the pro-domain of the human BDNF protein, is associated with increased susceptibility to a variety of brain disorders. As the met-BDNF fails to localize into secretory granules or synapses [23,24], it seems that this polymorphism affects BDNF-dependent regulation of synaptic transmission and plasticity [25]. Some studies on cells overexpressing BDNF suggest that un-cleaved proBDNF can be released via the regulated pathway [26], while other studies have shown detectable levels of proBDNF in some CNS areas including the amygdala and hippocampus [27,28]. Opposite to these findings, Matsumoto and colleagues [6] and Dieni and collaborators [29] found that in non-pathological conditions, proBDNF is rapidly converted intracellularly to mature BDNF, which is stored and released by excitatory inputs, while non detectable levels of proBDNF are released. Dieni and collaborators also determined that BDNF is 10-fold more abundant that proBDNF in the adult mouse brain, suggesting that most proBDNF molecules are cleaved in regulated secretory vesicles under non-pathological conditions. The most plausible explanation about the differences on the release or not of proBDNF could be related to the activity of the convertases, which also could be controlled by neuronal activity. Altogether, these findings highlight the importance of neuronal activity in the secretion of BDNF, as well as the enhancement of synaptic transmission and synaptic plasticity [25].

Once secreted, BDNF and proBDNF bind to two sets of receptors. First, BDNF binds with high affinity to one of the members of the tropomyosin receptor kinase family, the TrkB receptor. This is a transmembrane protein with tyrosine kinase activity that induces the activation of the Ras/Rap-MAPK, PI3K-AKT, and PLCg-PKC cascades after ligand-dependent dimerization. The BDNF-TrkB pathway is known to regulate and coordinate numerous cellular functions such as neuronal survival, growth, and differentiation of neurons during development, activity-dependent synaptic plasticity, as well as processes of memory and learning in the mature CNS [25,30,31]. On the other hand, proBDNF, as well as the other pro-neurotrophins and neurotrophins, binds to the p75NTR receptor, a member of the tumor necrosis factor receptor (TNFR) superfamily [32,33]. This single transmembrane domain receptor has at its N-terminal side a unique extracellular domain with four cysteine-rich repeat domains negatively charged facilitating the binding of the neurotrophins. At the C-terminal side it also contains one cytoplasmatic domain lacking catalytical activity but can promote the activation of the Jun kinase and NF-KB pathways and the transcription of multiple genes [33,34,35]. The receptor p75NTR plays an important role in survival of specific neuronal populations during development but can also mediate death of neural cells both in normal and pathological conditions [9,33,36,37,38]. Therefore, BDNF and proBDNF can differentially activate different receptors and molecular pathways and mediate opposite effects such as neuronal survival or death. While BDNF downregulation has been directly associated with brain neurological conditions such as depression, Huntington’s disease, and Rett syndrome, proBDNF secretion inhibits synapse formation, dendritic arborization, and impairs synaptic transmission and plasticity [39,40,41,42,43,44]. Interestingly, different neuronal populations require different types of neurotrophins, which exert long- and short-term actions. This indicates a temporal and spatial dependent mode of action of the neurotrophins during development and adulthood. For example, BDNF increases the arborization of hippocampal GABAergic cultured neurons but fails to do the same on the glutamatergic neurons from the same culture. Additionally, lack of BDNF in the CNS causes a reduction of the size of the GABAergic neurons of the striatum but not of the pyramidal hippocampal neurons [45,46,47]. Altogether, these data indicate that the differential response to BDNF for different neurons is part of the neuron-intrinsic program.

## 2. BDNF Expression

BDNF is the member of the family of neurotrophic factors most predominantly expressed in the postnatal mouse brain, although its expression has also been demonstrated in peripheral tissues such as lung, heart, thymus, liver, spleen, and also in platelets [48,49,50,51,52]. BDNF mRNA and protein levels are very low in developing regions of the CNS. Between the 11th and 12th day of rat and mouse embryogenesis (E11.5–E12.5) there is an increment in the BDNF expression levels, coinciding with the developmental onset of neurogenesis and an increment in neuronal activity [49,53]. Interestingly, BDNF expression in the newborn is most prominent in the CNS regions, in which neurogenesis has already occurred but at limiting amounts [29]. These low levels of BDNF cause a competition among neurons in PNS, which results in a normal loss of excess neurons [54]. BDNF mRNA levels increment until adulthood, when it becomes the most predominant neurotrophin expressed in the postnatal brain, with the highest levels in the neocortex, amygdala, hypothalamus, and the mossy fiber terminals at the hippocampus, where BDNF accumulates in presynaptic DCVs [29]. The relation between increment of mRNA BDNF expression and increment of the neuronal activity during brain development until adulthood is related. Indeed, it was shown that kainic acid-induced seizure activity caused an increment of BDNF mRNA levels in the hippocampus correlating the increment of neuronal activity with a transcriptional regulation of the BDNF gene [55].

One remarkable feature of the BDNF gene is its complex structure, with eleven different exons in humans, nine in rodents, and nine alternative promoters in both groups that control the cells and tissue where BDNF must be produced during development and adulthood. Despite this complexity, the coding sequence for the pre-proBDNF is located only in the exon IX, while all the other exons correspond to untranslated regions that regulate the complex spatial-temporal expression of BDNF in response to a great variety of stimuli [56,57,58]. For example, BDNF transcripts containing exons IV and IX are expressed at high levels in human heart, placenta, and prostate; transcripts containing exon VI are found in lung, kidney, muscles, and stomach, while transcripts containing exons II and VII are found to be exclusively expressed in the brain [58]. The presence of nine alternative promoters in the BDNF gene indicates that these could also be involved in developmental stage-specific expression and cell-type-specific expression, giving additional flexibility to the control of BDNF expression. For example, isoforms generated from promoters I and IV are found to be produced after activation of glutamate NMDA-type receptors in neurons, as well as after activation of the L-type voltage gated Ca^2+^ channels (L-VGCC) after neuronal depolarization. Promoter I contains at least one active cAMP and Ca^2+^-responsive element [59] that mediates BDNF transcription after binding of the neuronal-activity dependent transcriptional factor CREB (cAMP and Ca^2+^- response element-binding protein). Promoter IV can also mediate activity-dependent BDNF transcription as it contains three calcium-dependent responsive elements (CaRE) involved in regulating BDNF after activation of the NMDA receptors, which causes Ca^2+^ influx into the neurons and activation of the proteins Ca^2+^/calmodulin-dependent protein kinase II and IV [60]. This promoter also contains one binding site for the transcriptional factor NFAT (nuclear factor of activated T-cells) and one binding site for the negative regulatory element BHLHB2 (basic helix–loop–helix protein B2) [61,62]. While disruption of the BDNF promoter I impairs the response to neuronal depolarization [63] and enhances aggression in mice [64], disruption of promoter IV has been associated with impaired inhibitory neuronal activity in the prefrontal cortex, leading to depression-like behavior in mice [65].

Recent studies have shown that miRNAs can regulate BDNF expression and function. For example, miR-134 has been shown to mediate post-transcriptional regulation of CREB [66,67], while reduction in miR-202-3p and miR-124 levels causes upregulation of BDNF expression, while overexpression of miR-153 causes an increase in BDNF and the proliferation ability of hippocampal neurons [68,69,70]. There is also evidence suggesting a relationship between stress exposure and epigenetic regulations of BDNF with the development of psychiatric disorders. For example, changes in exon IV methylation levels are implicated in depression, while some antidepressants can promote BDNF transcription [71,72,73]. In addition, some drugs that can modulate neuronal activity are able to regulate BDNF expression. For example, the AMPA receptor agonist ampakine may help in correcting CNS dysfunction by increasing BDNF levels in animal models of Huntington’s disease and Rett syndrome, a rare autism spectrum disorder due to mutations in the DNA binding protein MeCP2 (methyl-CpG-binding protein 2) [74,75,76]. Moreover, administration of fingolimod, a sphingosine-1 phosphate analogue, effectively increases BDNF levels by inducing the activation of the TrkB-MAPK-CREB pathway in a mouse model of Rett syndrome [4]. Finally, BDNF levels are decreased in the brains of mice lacking MeCP2 and correlate with symptom progression [77], while BDNF can promote MeCP2 phosphorylation [78]. Together, these data clearly show that activity-dependent transcription plays important roles in BDNF transcription, indicating a clear and direct relationship between proBDNF/BDNF regulation and synapse development and plasticity both in normal and pathological conditions.

## 3. Regulated BDNF Secretion

To exert its functions BDNF needs to be secreted by the corresponding cell type at the right time and place. Researchers have used different methods to detect BDNF: ELISA measurements and immunostaining to detect the very low endogenous BDNF levels and live cell imaging for fluorescently labeled BDNF. However, the studies using BDNF tagged with GFP and other overexpression systems should be cautiously considered since the processing of pro-BDNF is affected [79] and may affect BDNF secretion that could be different from endogenous. Once synthesized, BDNF is stored in DCVs until it is released in response to cytoplasmic Ca^2+^ elevation, which works as a trigger for the process. Two sources can increase this Ca^2+^ concentration to elicit BDNF secretion: the extracellular Ca^2+^ and the intracellular Ca^2+^ stored in different organelles. Whereas glutamate-induced BDNF release from brain slices or primary neuronal cultures depends on Ca^2+^ release from internal Ca^2+^ stores, BDNF secretion in response to action potential is dependent on extracellular Ca^2+^ [80,81,82].

Among the cells that can secrete BDNF, neuronal cells are the most described; although, there are several reports indicating that astrocytes and microglia can also secrete BDNF. The timing of BDNF secretion is controlled by several different stimuli such as neuronal electrical activity including depolarization upon increasing extracellular K^+^-concentration, high frequency stimulation (HFS), or theta-burst stimulation (TBS), but also by neuropeptides (neurotrophins, CGRP, etc.) and different compounds (glutamate, ATP, capsaicin, adenosine, etc.). For a complete list check Table 1. In developing and mature neurons, the best characterized electrical stimuli that trigger BDNF release are prolonged depolarization, HFS, or TBS [83]. In electrically non-excitable cells, BDNF released stimuli differ from the major ones in neuronal cells. Thus, binding of extracellular nucleotides [84,85,86,87], proinflammatory factors [88], etc., to their corresponding receptors has been shown to stimulate BDNF secretion from astrocytes and microglia. 

Regarding the place of BDNF release from neurons, it has been reported that it could be postsynaptic [99,114,115,116,117,118] and presynaptic [119,120,121,122,123,124,125] but also somatic and dendritic [99,114,115,116,119,122,126]. In addition, the synthesized BDNF could be stored and secreted from DCVs or endosomes previously endocytosed and accumulated close to the cell membrane of neurons and astrocytes [82,111,127,128].

## 4. BDNF Release during Development

The onset of BDNF expression in CNS is different in the rodent hippocampus and cortex; whereas BDNF expression becomes apparent at E15.5 in the hippocampus, and in cortical neurons it appears at P4 [51,129]. BDNF secretion depends on the stage of the neuronal networks since the electrical activity patterns required to elicit BDNF release are different in immature and mature synaptic circuits. In developing hippocampal and cortical networks three different electrical activity patterns have been described to be relevant to form synaptic circuitry: brief L-type VGCC-mediated spikes, synchronous plateau assemblies and giant depolarization potentials (GDPs) [130,131,132]. All these electrical activity patterns depend on the activation of VGCC, and, therefore, BDNF release is dependent on Ca^2+^ entry through these channels [116,122,126,133].

## 5. BDNF Release from Mature Neurons in Synaptic Plasticity

Synaptic plasticity, such as long-term potentiation (LTP) and long-term depression (LTD), is thought to be a cellular model of learning and memory processes. The best-known patterns to induce LTP of synaptic transmission in the hippocampus are HFS, TBS, or pairing protocols such as spike timing-dependent plasticity (STDP). Protocols that elicit LTD of synaptic efficacy are characterized by low-frequency synaptic stimulation (LFS). Endogenous BDNF release has been intensively studied as an activity-dependent mechanism of LTP in brain slices and in neuronal cultures from different brain regions such as PNS-cultured neurons [134], hippocampal cultures [99,115,122,135], amygdala [136], and cortico-striatal synapses [137]. However, in dorsal horn slices from the lumbar spinal cord the above patterns of high-frequency stimulation did not induce efficient release of BDNF [93]. In this case effective stimuli to induce release of BDNF was TBS and capsaicin injection in the hind paw, which in turn produces bursting activity reminiscent of TBS in nociceptors [93]. 

In addition to the relevance of newly synthesized BDNF on LTP, release of BDNF has been also described by cells that do not express but accumulate it. This recycled BDNF is previously secreted by other expressing cells and then endocytosed and stored close to the plasma membrane in endosomes. This re-exocytosis of BDNF has been reported for cultures of mature neurons and astrocytes [82,87,111,138]. The relevance of this mechanism has been revealed in connected neuronal circuits and surrounding astrocytes, as re-use of endocytosed BDNF enables the replenishment of the BDNF pool for late-LTP [82] and long-distance distribution of synthetized BDNF across synaptically connected neuronal circuits and surrounding astrocytes [127,139,140]. Independently of BDNF source, its role controlling LTP is unquestionable.

## 6. BDNF Release from Astrocytes

Aside from the controversy whether proBDNF is secreted or not [6,26,27,29], it has been reported that proBDNF is endocytosed by astrocytes during TBS-LTP in hippocampal or perirhinal cortex slices in a p75-dependent manner [111,112]. This event is prevented by the cleavage of proBDNF by plasmin or by the deletion of glial p75 neurotrophin receptor [111,112]. Endocytosed proBDNF is then intracellularly cleaved, and the newly mature BDNF contributes to the maintenance of TBS-induced LTP in a postsynaptic TrkB-dependent manner [112]. Among the stimuli inducing release of recycled BDNF from astrocytes, glutamate release from the presynaptic terminal during TBS is very relevant [82,95,112]. In addition, elevated extracellular potassium concentration upon TBS activity [87], agonists of AMPA or mGluRI/II receptors [111], ATP acting through receptor P2X7 [91], and BDNF itself [108,141] can elicit release of recycled BDNF from astrocytes, which acts during synaptic plasticity. Another potential mechanism of BDNF re-endocytosis in hippocampal-derived astrocytes is through TrkB.T1. This truncated receptor, which lacks the kinase domain, is highly expressed in astrocytes and mediates the storage of endocytosed BDNF, which will be ready to be secreted in response to different stimuli [142].

Astrocytes also play an important role in immunoregulatory functions through different compounds such as the prostaglandin E2 and tumor necrosis factor-alpha, which have been shown to induce release of BDNF [109,113,143]. It has been reported that in in vitro cultured astrocytes, BDNF secreted from these glial cells protects dopaminergic neurons in response to oxidative stress induced by 6-hydroxydopamine [98]. Therefore, BDNF release may play relevant functions of astrocytes.

## 7. BDNF Release from Microglia

Microglia, the primary immune cells of the brain, also play an important role in neuroprotection. BDNF expression and secretion have been reported for both activated M1 and M2 phenotypes but also for non-activated M0 microglia [84,101,107,144]. Microglia polarization to M1 phenotype by lipopolysaccharide (LPS) and to M2 by IL-4 induces extracellular accumulation of mature BDNF in microglial cultures [101,106,107]. Microglia cells play a very relevant role in nociception, since expression of the surface P2X4 adenosine receptors (P2X4R) are associated with central sensitization such as pain hypersensitivity or allodynia [86,145]. The mechanism seems to involve the binding of extracellular ATP to P2X4R, which increases intracellular Ca^2+^ levels and through the small G protein MAPK/p38 leads to BDNF release [84,85,90,92].

## 8. Molecular Machinery Involved in BDNF Release

BDNF secretion is tightly controlled by different stimuli but also by different transmembrane and intracellular proteins that are implicated in raising cytoplasmic Ca^2+^ levels and responding to this elevation. In the following sections, we will describe most of the known proteins (Table 2).

### 8.1. Trk Neurotrophin Receptors

Neurotrophins bind to their corresponding Trk receptor, nerve growth factor (NGF) to TrkA, BDNF and neurotrophin-4 (NT-4) to TrkB, and neurotrophin-3 (NT-3) to TrkC [30,161]. The role of neurotrophins through the activation of Trk receptors in the regulation of BDNF release was described in two pioneer studies carried out in primary neurons and PC12 cells [104,108]. This secretion is dependent on Ca^2+^ release from intracellular stores [108] and phospholipase C (PLC) [80]. Other authors have observed that treatment of cortical/hippocampal slices and dorsal root ganglion neurons with NT-3 or NT-4 and NGF, respectively, led to BDNF release [105]. In addition, the role of TrkB.T1 receptor in the release of previously internalized BDNF from astrocytes has been described [142]. *TrkB.T1* knockout mice show that the truncated receptor participates in several neurological disorders such amyotrophic lateral sclerosis, Alzheimer’s disease, and Parkinson´s disease in which BDNF levels are also altered (reviewed by Tessarollo and Yanpallewar, 2022) [162]. These studies point out that neurotrophin receptor-regulated BDNF secretion may provide a positive feedback mechanism for physiological functions such as selective stabilization of synaptic connections, potentiation of synaptic transmission, and memory formation. However, an impairment of BDNF availability is observed in different diseases when neurotrophin receptors are altered.

### 8.2. ARMS/Kidins220

ARMS/Kidins220 (ARMS hereinafter) is a transmembrane scaffold protein with pleiotropic functions in the nervous system and other systems [163]. The gene coding for ARMS was cloned independently as a protein kinase D (PKD) substrate [164] and as a downstream target of the signaling mediated by neurotrophins and ephrins [165]. ARMS structure is composed of eleven ankyrin repeats, a Walker A and B domain, four transmembrane regions, a proline-rich domain, a sterile-alfa motif (SAM), a kinesin light chain (KLC)-interacting motif (KIM), and a PDZ-binding motif. ARMS interacts with multiple proteins within its multiple domains acting as a signaling platform [163]. 

The first article implicating ARMS in secretion was related with neurotensin release downstream of PKD [166]. The authors reported a positive role of ARMS in neurotensin secretion from BON cells downstream of PKD signaling pathway in response to phorbol esters, in which ARMS seems to regulate neurotensin vesicle transport to the plasma membrane [166]. The involvement of ARMS in secretion in the nervous system comes from the regulated secretion in the PC12 neuronal cell line in response to NGF [167]. However, in this case it was reported that ARMS plays a negative role, since reducing and increasing its levels resulted in increased and decreased secretion, respectively. Mechanistically, ARMS was acting together with synembryn, a protein previously involved in the neurotransmitter release at the neuromuscular junction in *C. elegans* [168], and upstream of Gαq, Trio, and Rac proteins [168]. Interestingly, proliferating PC12 cells showed high levels of ARMS with very low NGF-mediated secretion, whereas differentiated PC12 cells showed low levels of ARMS and high NGF-mediated secretion, and manipulation of ARMS levels demonstrated its negative effect on release [167]. Furthermore, the role of ARMS in secretion was studied in CNS and PNS in two different studies looking at regulated BDNF secretion. In the first study, López-Benito and collaborators demonstrated that ARMS negatively regulates BDNF release in response to NT-3, NT-4, and depolarization in vitro in cultured cortical neurons [105]. As a result of knocking down ARMS levels in vivo both in the cortex and hippocampus, there is an accumulation of BDNF in the striatum, a region that does not express BDNF but that receives it from the cortex and hippocampus. Mechanistically, the ARMS effect on BDNF release depends on the regulation of Syt4 levels [105], a protein which has been previously observed to modulate BDNF secretion (see below) [119]. In the second study, knocking down ARMS in TrkA-expressing cells resulted in an enhanced release of BDNF from dorsal root ganglion neurons in the spinal cord in response to capsaicin injection [94]. As a result, ARMS modulates thermal and inflammatory nociception, effects of which directly depend on the presence of BDNF. Importantly, ARMS protein levels are downregulated by noxious stimuli, which trigger neuronal activity [94]. Altogether, these studies point to a seminal role of ARMS on functions directly dependent on BDNF secretion.

The relevance of ARMS controlling BDNF release is palpable in physiological and pathological conditions. It is demonstrated that ARMS influences synaptic activity by controlling basal synaptic transmission [169] and enhances LTP in heterozygous *ARMS* mice [170]. Since LTP depends on BDNF secretion [171,172], strict control of ARMS protein quantity is required. ARMS levels are regulated by calpains [170,173], which are a family of evolutionarily conserved Ca^2+^-dependent cysteine proteases that function in numerous processes including synaptic plasticity and neuronal survival/degeneration [174,175]. The effect on LTP is linked to calpain-mediated cleavage of ARMS [170], a phenomenon occurring in response to neuronal activity. Calpain-dependent proteolysis is ideal for ARMS degradation because its enzymatic activity is induced by increases in intracellular Ca^2+^ levels in response to neuronal activity. Since ARMS regulates BDNF secretion, LTP induction, and modulates functions of GluA1 and NMDARs [169,176], the regulation on ARMS levels has a direct impact on neuronal maturation and synaptic plasticity [169,170,177]. It has been reported that ARMS is also involved in the development of brain pathological conditions. For example, it has been observed that in two different mouse models of Huntington’s disease (HD) increased levels of ARMS caused a deficit in the regulated secretion of BDNF [105]. In addition, the hippocampus and prefrontal cortex (PFC) of HD patients [105] and the temporal cortex of Alzheimer´s Disease (AD) patients [178] display elevated ARMS levels. Altogether, these studies suggest that ARMS is a key protein in the regulation of the secretion of BDNF both in physiological and pathological conditions in the nervous system.

### 8.3. PKG

Protein kinase G (PKG) isozymes belong to the family of serine/threonine kinases that are activated by cGMP and are homologous to cAMP-dependent-protein kinase A (PKA) [179]. The involvement of PKG in BDNF secretion was originally reported in cultured hippocampal neurons expressing exogenous BDNF [151]. In response to nitric oxide (NO), cGMP levels are raised activating PKG, which in turn prevents Ca^2+^ release from inositol 1,4,5-triphosphte-sensitive stores leading to a rapid down-regulation of BDNF secretion [151]. Another report indicates that exogenous NO abolishes BDNF release from in vitro cultures of newborn rat nodose ganglion neurons stimulated with single electrical pulses, but this effect seems not to involve PKG [152]. Recently, it has been reported that in spinal terminals of nociceptors, presynaptic NMDARs activation in response to tissue inflammation enhances BDNF secretion. This effect is dependent on prolonged Ca^2+^ elevation and PKG activation leading to synaptic potentiation in the inflammatory state [153]. Although PKG activation evokes opposite effects of BDNF secretion in different systems, PKG signaling pathway represents a signaling mechanism by which neurons can modulate BDNF secretion.

### 8.4. Rab3a-Rim1

Rab3 is a small GTPase localized on membranes of DCVs [180]. This protein acts together with its neuronal effector, Rab3 interacting molecule (Rim1), in the release of neuropeptides and neurotrophins [155]. In astrocytes, Rab3a is involved in the docking of BDNF vesicles on the plasma membrane, which is impaired by mutant huntingtin [154], whereas in neurons, DCV exocytosis is undetectable upon RIM1/2 deletion [155]. Rab3 plays a crucial role in the presynaptic, but not postsynaptic, component of BDNF-induced synaptic charge [156,157], an effect that requires Rab3/Rim1 to activate proline-directed Ser/Thr protein kinases [96]. Altogether, these studies indicate that Rab3a participates in docking DCVs containing BDNF, which participate on synaptic transmission.

### 8.5. Munc18

Munc18-1 is a member of the Sec1/Munc18 (SM) family of proteins playing fundamental roles in membrane trafficking [181]. It has been implicated in the synaptic vesicle docking, priming, and fusion, functions that depend, at least in part, on its capacity to bind the neuronal SNAREs [182]. *Munc18-1* knockout mice die at birth and show massive neurodegeneration [183]. Munc18-1-deficient hippocampal neurons displayed reduced BDNF secretion [148], and BDNF application on these neurons delays their death and rescues their severe synaptic dysfunctions [148,149,184]. The impaired neuropeptide secretion may explain aspects of the behavioral and neurodevelopmental phenotypes that were observed in *Munc18-1* heterozygous mice [149]. Moreover, BDNF/TrkB signaling in response to synaptic activity at the neuromuscular junction prevents Munc18-1 phosphorylation, preventing its binding to syntaxin [185]. All together, these data suggest a role of Munc18-1 in BDNF secretion and the presence of a positive feedback loop between BDNF/TrkB signaling and Munc18-1 function.

### 8.6. CAPS2 

Ca^2+^-dependent activator protein for secretion (CAPS2) is an 1803 amino acid protein that has structural features including a dynactin1 interaction domain (DID), a C2 domain, a PH domain, and the Munc13-1-homologous domain (MHD) containing a syntaxin-interacting domain. CAPS2 shares 70.4% amino acid identity with CAPS1 and have different splice variants [110,146]. CAPS2 is enriched in vesicular structures of the presynaptic parallel fiber terminals of cerebellar granule cells mediating the depolarization-dependent release of NT-3 and BDNF that regulates cell differentiation and survival during cerebellar development [110,146]. In addition, CAPS2 expression in hippocampal GABAergic neurons regulates BDNF secretion, development of hippocampal GABAergic neurons and their synapses without affecting excitatory hippocampal neurons [125]. *CAPS2* knockout mice exhibited impaired activity-dependent BDNF secretion, reduced late-phase long-term potentiation at CA3–CA1 synapses, decreased hippocampal theta oscillation frequency, and increased anxiety-like behavior [110,125,147]. In addition, *CAPS2* knockout mice show autistic-like cellular and behavioral phenotypes, and, in autistic patients, an aberrant alternatively spliced CAPS2 mRNA lacking the exon 3 was identified [110]. This exon codes for a region involved in the binding to dynactin 1 motor protein and proper localization of CAPS2 in axons [110]. Mice with the exon 3 deleted showed a reduction in BDNF release from axons and autistic-like behavior [123]. The deficits in CAPS2, or its splicing variant, in the regulation of BDNF secretion may be responsible for the development and maturation of synapses and in the balance between the inhibitory and excitatory systems. Altogether, CAPS2 seems to be responsible for BDNF secretion, and dysregulation of the protein provokes deficits in brain development and autism-related behaviors in mice and patients.

### 8.7. Synaptotagmin 4 and Synaptotagmin 6

Synaptotagmin 4 (Syt4) belongs to the synaptotagmin family of proteins [186]. Syt4 harbors an aspartate-to-serine substitution in a Ca^2+^ coordination site of the C2A domain and, therefore, is unable to bind Ca^2+^. Syt4 binds to SNARE proteins but fails to bind more avidly to SNAREs or to penetrate membranes in response to Ca^2+^. Thus, Syt4 can join the fusion complex but prevents an essential fusion step limiting exocytosis [181]. Syt4 is mainly expressed in brain and neuroendocrine tissues at relatively low levels, but it is induced by neuronal depolarization, seizures, and psychoactive drugs [187,188,189,190,191]. Syt4 is localized to BDNF-containing vesicles in hippocampal neurons, and its role inhibiting BDNF release at both axons and dendrites was demonstrated using knockout and overexpressing hippocampal neurons [119]. This effect was at the postsynaptic site of BDNF release affecting indirectly at the rate of synaptic vesicle exocytosis from presynaptic terminals. In addition, *Syt4* knockout mice showed enhanced TBS-mediated LTP, which depended entirely on disinhibition of BDNF release [119]. Furthermore, Syt4 seems to work together with ARMS since depletion of ARMS levels abolished the inhibitory effect of Syt4 overexpression on BDNF secretion [105]. Therefore, Syt4 seems to be instrumental for the control of BDNF release and LTP induction.

It is known that hippocampal neurons could recycle BDNF after endocytosis for activity-dependent secretion, and this BDNF could replace its new synthesis required for the late phase of LTP [82]. Wong and collaborators reported that another member of the synaptotagmin family, synaptotagmin 6 (Syt6), promotes activity-dependent exocytosis of BDNF-containing endosomes [159]. Specific down-regulation of Syt6 abolished activity-driven release of endocytosed BDNF at postsynaptic sites [159]. In addition, complexin1, which activates and clamps vesicular exocytosis interacting with SNARE proteins, is required for activity-dependent exocytosis of BDNF at postsynaptic sites, but at the same time it prevents spontaneous exocytosis of docked endosomes [159]. Additionally, the protein Vamp3 has been recently implicated in BDNF release from endocytosed BDNF [160]. Thus, using different proteins, endocytosed BDNF can be stored with the activity-dependent releasable pool required for LTP maintenance.

### 8.8. SNARE Proteins

SNARE complexes are formed primarily by synaptobrevin/VAMP2 (Syb2), SNAP25, and syntaxin1 and are seminal for presynaptic release of synaptic neurotransmitter vesicles overcoming the energy barrier for their fusion with the plasma membrane [183]. In addition, Shimojo and collaborators found that Syb2 and SNAP25 are involved in BDNF release in cortical neurons together with SNAP47 [158]. Previously, it was reported that Syb2 and SNAP25 colocalized with CAPS2 [192]. Thus, SNARE protein complexes also participate in BDNF secretion.

### 8.9. mGluR

The metabotropic glutamate receptor (mGluR) family is composed of three groups based on the sequence homology and G-protein coupling, correspondingly mGluR1 and 5 (Group I), mGluR2 and 3 (Group II), and Group III, including mGluRs 4, 6, 7, and 8. [193]. The main intracellular transduction cascade activated by group I mGluRs, coupled to Gq/G_11_, is the phospholipase C pathway. This activation results in the generation of inositol 1,4,5-trisphosphate (IP3) and Ca^2+^ mobilization from intracellular stores [194]. mGluRI-dependent activation of PLC induces BDNF release in hippocampal neurons [80], in astrocytes [95] and in oligodendrocytes [97].

## 9. Conclusions

BDNF plays many pleiotropic functions in different biological systems. Since its discovery, BDNF has been implicated in survival and differentiation of various neuronal populations, modulation of synaptic activity, expression of genes required for long-term effects, and even on the survival of cancer cells through anoikis. Dysregulation of BDNF functions have many implications in terms of diseases such as obesity, anxiety, depression, and neurodegenerative diseases. Several researchers and companies have tried to generate TrkB agonists as therapeutic agents to mimic BDNF functions, but this has been an unaccomplished challenge due to the spatio-temporal complexity of BDNF regulation. Therefore, BDNF expression, location, and secretion need to be finely tuned to properly accomplish such complex duties. The thorough knowledge of these processes will provide insights to develop new molecules and treatments for the associated diseases with BDNF.

## Figures and Tables

**Table 1 biomolecules-13-00789-t001:** Stimuli that lead to BDNF secretion.

Stimulus	Cell Type	Proteins Involved	References
**Adenosine**	Microglial cells	GPCR, PKA, PLC	[89]
**ATP**	Astrocytes/Microglial cells	P2X4, Ras, p38-MAPK	[80,84,85,87,90,91,92]
**Capsaicin**	Neurons	TRPV1	[93,94]
**Glutamate**	Neurons/ Astrocytes/ Oligodendrocytes	NMDARs, AMPARs, mGluRs	[80,95,96,97]
**5-hydroxytryptamine (5-HT; serotonin)**	Cultured NG2 glial cells	5-HTR	[98]
**High frequency stimulation (HFS)**	Neuron	NMDARs, AMPARs, mGluRs	[99,100,101]
**Ketamine**	Astrocytes	NMDARs	[102,103]
**KCl**	Neurons/Astrocytes	VGCC	[80,104,105]
**Lipopolysaccharide** **Ceramide**	Microglia	PKC	[106,107]
**NGF**	Neurons	TrkA	[105,108]
**NT-3**	Neurons	TrkB, TrkC	[105,108]
**NT-4**	Neurons	TrkB	[105,108]
**Prostaglandine E2**	Astrocytes/Microglial cells	GPCR, PKA, PLC	[109]
**Rhythmic neuronal discharges**	Neuron	NMDAR, AMPARs, mGluRs	[10]
**Theta-burst stimulation (TBS)**	Neuron	NMDARs, AMPARs, mGluRs	[82,83,93,95,110,111,112]
**TNF**	Astrocytes	TNFR	[113]

**Table 2 biomolecules-13-00789-t002:** Proteins involved in BDNF secretion.

Protein	Effect	Cells	BDNF Source	References
**ARMS/Kidins220**	−	Neurons	synthesized	[94,105]
**CAPS2**	+	Neurons	synthesized	[110,125,146,147]
**mGluR**	+	Neurons/Astrocytes	endocytosed/recycled	[80,95,97]
**Munc-18**	+	Neurons	synthesized	[148,149]
**PKA**	+	Neurons/Glial cells		[150]
**PKG**	−/+	Neurons		[151,152,153]
**PLC**	–/+	Neurons/Astrocytes	synthesized endocytosed/recycled	[80,95,97]
**Rab3a/Rim1**	+	Neurons/Astrocytes	synthesized endocytosed/recycled	[96,154,155,156,157]
**SNAP25**	+	Neurons	synthesized	[158]
**SNAP47**	+	Neurons	synthesized	[158]
**Synaptobrevin2**	+	Neurons	synthesized endocytosed/recycled	[158]
**Synaptotagmin4**	–	Neurons	synthesized	[105,119,159]
**Synaptotagmin6 /complexin 1**	+	Neurons	endocytosed/recycled	[159]
**Trk receptors**	+	Neurons	synthesized	[104,105,108]
**Vamp3**	+	Astrocytes	endocytosed/recycled	[160]

## Data Availability

No new data were created or analyzed in this study. Data sharing is not applicable to this article.

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
