# Peer review of "Mechanisms Controlling the Expression and Secretion of BDNF"

_biomolecules, 2023, doi:10.3390/biom13050789_

Round 1

Reviewer 1 Report

This a courageous attempt by highly knowledgeable scientists to summarise work by a number of laboratoriess attempting to dissect molecular mechanisms involved in the secretion of BDNF. The review also deals with the regulation of expression of the gene, an endeavour that is less challenging and an area that has been covered in previous reviews. Regarding mechanisms possibly regulating the secretion of BDNF, I would suggest cautioning that this type of work is exceedingly difficult given the very low concentrations of endogenous BDNF. In addition, much of the work has been done in vitro, with some of the references cited utilizing cell lines. Problematic is also the fact that many of the studies (see for example Ref. 85, 86, 87, 88, 93, 95) deal with overexpression of neurotrophin cDNAs, including in particular BDNF constructs tagged with reporters such as GFP.  The use such constructs greatly complicates the interpretation of the results. Indeed, it would appear a priori quite improbable that such comparatively long tags would not affect key aspects of the biochemistry and cell biology of BDNF. In the vast majority of cases, these tools have been insufficiently validated if at all and it turns that, as could be expected, the GFP tag interferes with the processing of pro-BDNF as well as with the secretion of BDNF in transfected cells (see for example Fig. 1 in PMID: 31882533). Mice engineered to express constructs translating BDNF covalently linked with GFP turned out either not to be viable whereby sections of the brains of survivors indicated a distribution of the GFP signal that does not reflect the distribution of endogenous BDNF, presumably because the cleavage of the tag (see PMID 30929164). Last and along similar lines, the authors may want to add a note of caution when discussing the distribution of BDNF in non-neuronal cells. Most studies referring to astrocytes involve cultured astrocytes, known to have different expression profiles in vitro compared with astrocytes in vivo. In view of this, the statement that “relevant functions of astrocytes depend on their BDNF release” (line 248) may be reconsidered. These reservations also apply to microglial cells and a recent study has questioned the presence of BDNF in microglial cells (PMID: 36726454).

I also have specific recommendations:

1.      When discussing the cleavage of pro-BDNF (Line 36) the author should add reference (PMID: 24101515) as it identifies PC7 as the convertase involved in the ENDOGENOUS cleavage of pro-BDNF

2.      Line 107-109 “That the low levels of BDNF cause a competition among neurons, what results in a normal loss of excess neurons” only holds true for PNS, not CNS neurons.

3.      Some sentences are difficult to understand:
Line 16 “how changes of their levels or performance in these proteins”
Line 32-33 replace “suffers” by “undergoes”

Line 114 “is not casual”

Line 224 That platelets may release BDNF following uptake (from whatever source) should be documented if this is what the authors mean.

Author Response

Reviewer 1

This a courageous attempt by highly knowledgeable scientists to summarise work by a number of laboratoriess attempting to dissect molecular mechanisms involved in the secretion of BDNF. The review also deals with the regulation of expression of the gene, an endeavour that is less challenging and an area that has been covered in previous reviews. Regarding mechanisms possibly regulating the secretion of BDNF, I would suggest cautioning that this type of work is exceedingly difficult given the very low concentrations of endogenous BDNF. In addition, much of the work has been done in vitro, with some of the references cited utilizing cell lines. Problematic is also the fact that many of the studies (see for example Ref. 85, 86, 87, 88, 93, 95) deal with overexpression of neurotrophin cDNAs, including in particular BDNF constructs tagged with reporters such as GFP.  The use such constructs greatly complicates the interpretation of the results. Indeed, it would appear a priori quite improbable that such comparatively long tags would not affect key aspects of the biochemistry and cell biology of BDNF. In the vast majority of cases, these tools have been insufficiently validated if at all and it turns that, as could be expected, the GFP tag interferes with the processing of pro-BDNF as well as with the secretion of BDNF in transfected cells (see for example Fig. 1 in PMID: 31882533). Mice engineered to express constructs translating BDNF covalently linked with GFP turned out either not to be viable whereby sections of the brains of survivors indicated a distribution of the GFP signal that does not reflect the distribution of endogenous BDNF, presumably because the cleavage of the tag (see PMID 30929164). Last and along similar lines, the authors may want to add a note of caution when discussing the distribution of BDNF in non-neuronal cells. Most studies referring to astrocytes involve cultured astrocytes, known to have different expression profiles in vitro compared with astrocytes in vivo. In view of this, the statement that “relevant functions of astrocytes depend on their BDNF release” (line 248) may be reconsidered. These reservations also apply to microglial cells and a recent study has questioned the presence of BDNF in microglial cells (PMID: 36726454).

We appreciate the positive comments of the reviewer.

We have included different statements to address the reviewer concerns: 1) related with the very low concentrations of BDNF (page 4 line 188 of the revised manuscript), and 2) related with fusions GFP-BDNF (page 4 lines 189-192 of the revised manuscript).

Finally, as suggested by the reviewer, we have modified the statement “relevant functions of astrocytes depend on their BDNF release” in the revised manuscript. This is now presented on page 6 lines 277-278 of the revised manuscript.

I also have specific recommendations:

  1. When discussing the cleavage of pro-BDNF (Line 36) the author should add reference (PMID: 24101515) as it identifies PC7 as the convertase involved in the ENDOGENOUS cleavage of pro-BDNF.

We have included the PC7 convertase and the reference.

This is now presented on page 1 line 36 of the revised manuscript.

  1. Line 107-109 “That the low levels of BDNF cause a competition among neurons, what results in a normal loss of excess neurons” only holds true for PNS, not CNS neurons.

We have modified the statement.

This is now presented on page 2 line 119 of the revised manuscript.

  1. Some sentences are difficult to understand:

Line 16 “how changes of their levels or performance in these proteins”
Line 32-33 replace “suffers” by “undergoes”

Line 114 “is not casual”

We appreciate the comments of the reviewer and have made the changes, which are now presented on lines 16, 32-33 and 124, respectively, of the revised manuscript.

  1. Line 224 That platelets may release BDNF following uptake (from whatever source) should be documented if this is what the authors mean.

We have removed platelets from the sentence.

This is now presented on page 6 line 250 of the revised manuscript.

Reviewer 2 Report

This is a nice and very comprehensive review of the pathways and mechanisms that regulate BDNF secretion. Overall, the authors have extensively and fairly cited the literature to support their narrative. My only suggestion would be to add in the concluding paragraph that the spatio-temporal complexity of BDNF secretion may pose challenges to use agonists to mimic BDNF activities. Moreover, it may be relevant to mention that selective TrkB agonists have been put forward as potential therapeutics (e.g. PMCID: PMC2823863PMCID: PMC2860903) but there are significant doubts on the efficacy and specificity of these compounds (e.g. PMID: 28831019PMCID: PMC3913682, https://www.science.org/content/blog-post/those-compounds-aren-t-you-think-they). The effects observed in animal studies could be due to mechanisms independent of direct TrkB activation. I think that this addition would be important also to explain that potential TrkB agonists would have limited efficacy because of the complex regulation of secretion described in the review. One minimal suggestion would be include the effect of truncated TrkB on the regulation of BDNF secretion. There is a significant amount of literature on the topic (reviewed in PMCID: PMC8934854). This could be easily done in the paragraph about TrkB (page 7).

Minor comments and format suggestions:

Line 33. “what promotes” should be “which promotes”.

Line 44. “Opposite to…” should be “Contrary to…”

Line 55. The references are not formatted.

Line 77. “shows” should be “has”

Line 79-80. “At the C-terminal side it also contains one cytoplasmatic domain that lacks any catalytical activity, despite which it can promote the activation of..” could be, “At the C-terminal side it has a cytoplasmatic domain lacking catalytical activity, but can promote the activation of…”

Line 101-102. “thymus and liver spleen and non-neural cells such as platelets”, could be “thymus, liver and spleen, and also in platelets”

Line 115. “…in the hippocampus, what correlates….” Should be “ in the hippocampus correlating…”

Line 156. “ due to mutations on the ..” should be “due to mutations in the “

Line 263-4. The sentence seems missing something…”we will describe most of the known.”

Table 2.  CAPS2 is indicated as Effect – but it appears as + from the text; Also, PKA is missing the +

Author Response

Reviewer 2

This is a nice and very comprehensive review of the pathways and mechanisms that regulate BDNF secretion. Overall, the authors have extensively and fairly cited the literature to support their narrative. My only suggestion would be to add in the concluding paragraph that the spatio-temporal complexity of BDNF secretion may pose challenges to use agonists to mimic BDNF activities. Moreover, it may be relevant to mention that selective TrkB agonists have been put forward as potential therapeutics (e.g. PMCID: PMC2823863; PMCID: PMC2860903) but there are significant doubts on the efficacy and specificity of these compounds (e.g. PMID: 28831019PMCID: PMC3913682, https://www.science.org/content/blog-post/those-compounds-aren-t-you-think-they). The effects observed in animal studies could be due to mechanisms independent of direct TrkB activation. I think that this addition would be important also to explain that potential TrkB agonists would have limited efficacy because of the complex regulation of secretion described in the review. One minimal suggestion would be included the effect of truncated TrkB on the regulation of BDNF secretion. There is a significant amount of literature on the topic (reviewed in PMCID: PMC8934854). This could be easily done in the paragraph about TrkB (page 7).

We appreciate the positive comments of the reviewer.

We have modified the concluding paragraph as suggested. We hope that it now complies with the reviewer´s expectations and it is now on page 11 lines 517-519 of the revised manuscript.

Finally, we have included the role of truncated TrkB receptor on the secretion and endocytosis of BDNF in astrocytes on page 6 lines 268-272 and page 7 lines 319-324.

Minor comments and format suggestions:

Line 33. “what promotes” should be “which promotes”.

Change has been made and it is now on page 1 line 33.

Line 44. “Opposite to…” should be “Contrary to…”

Change has been made and it is now on page 1 line 44.

Line 55. The references are not formatted.

References are now formatted and are now on page 2 line 62 of the revised manuscript.

Line 77. “shows” should be “has”

Change has been made and it is now on page 2 line 83 of the revised manuscript.

Line 79-80. “At the C-terminal side it also contains one cytoplasmatic domain that lacks any catalytical activity, despite which it can promote the activation of..” could be, “At the C-terminal side it has a cytoplasmatic domain lacking catalytical activity, but can promote the activation of…”

Change has been made and it is now on page 2 lines 85-86 of the revised manuscript.

Line 101-102. “thymus and liver spleen and non-neural cells such as platelets”, could be “thymus, liver and spleen, and also in platelets”

Change has been made and it is now on page 3 line 117 of the revised manuscript.

Line 115. “…in the hippocampus, what correlates….” Should be “ in the hippocampus correlating…”

Change has been made and it is now on page 3 line 126 of the revised manuscript.

Line 156. “ due to mutations on the ..” should be “due to mutations in the “

Change has been made and it is now on page 2 line 175 of the revised manuscript.

Line 263-4. The sentence seems missing something…”we will describe most of the known.”

We have modified the sentence and now reads as “we will describe most of the known proteins (Table 2)”.

This is now included in page 7 lines 306-307 of the revised manuscript.

Table 2.  CAPS2 is indicated as Effect – but it appears as + from the text; Also, PKA is missing the +

We apologize for the mistake regarding CAPS2. Now is corrected and the + for PKA is included.

This is now included in new Table 2 of the revised manuscript.

Reviewer 3 Report

The revision by Arévalo and Deogracias on BDNF secretion covers a very relevant topic and, for most of the time, it reads well. Some minor points should be revised before its publication.

The introduction is rather long and it covers many different topics (sorting, release of ProBDNF vs BDNF, BDNF receptors, cell type-specific function, developmental expression, transcriptional regulation, etc.). I recommend to make the introduction shorter, and perhaps to include a new section where these topics could be covered in different subsections.

The section on BDNF polymorphism Val66Met looks a bit out of context. I would try to place it in a different section or give it more context.

In the sentence 75 authors implied that proBDNF can also bind to TrKB, by saying: “can also bind to…” . This should be clarified.

Sentences: 32-34, 37-39, 39-41, 51-52, 285-287 don’t read well, and they should be rewritten.

Sentence is 453-456 in the conclusion is too long and it does not read that well.

Typos: There also evidences (line 149), fed back (line 433) should be corrected

Sentence in line 263-264 does not seem to be finished.

C. elegans in line 301 should be italicized.  

Author Response

The revision by Arévalo and Deogracias on BDNF secretion covers a very relevant topic and, for most of the time, it reads well. Some minor points should be revised before its publication.

We appreciate the positive comments of the reviewer.

The introduction is rather long and it covers many different topics (sorting, release of ProBDNF vs BDNF, BDNF receptors, cell type-specific function, developmental expression, transcriptional regulation, etc.). I recommend to make the introduction shorter, and perhaps to include a new section where these topics could be covered in different subsections.

We decided to include a long introduction to show the many aspects and complexity of BDNF to put in context that its secretion is quite complex and tightly regulated and, therefore, it can be affected by several different factors and situations.

The section on BDNF polymorphism Val66Met looks a bit out of context. I would try to place it in a different section or give it more context.

We have modified the text giving more context.

This is now included in pages 1-2 lines 46-55 of the revised manuscript.

In the sentence 75 authors implied that proBDNF can also bind to TrKB, by saying: “can also bind to…” . This should be clarified.

We have modified the statement and it is now presented on page 2 lines 80-82.

Sentences: 32-34, 37-39, 39-41, 51-52, 285-287 don’t read well, and they should be rewritten.

We appreciate the input of the reviewer. Each sentence has been rewritten and are now presented on lines 32-34, 38-39, 46-55 and 331-333, respectively.

Sentence is 453-456 in the conclusion is too long and it does not read that well.

We have rewritten this sentence that is now presented on page 11 lines 512-515 of the revised manuscript.

Typos: There also evidences (line 149), fed back (line 433) should be corrected

We have made these corrections that are now presented on page 3 line 160 and on page 10 lines 489-490 of the revised manuscript.

Sentence in line 263-264 does not seem to be finished.

We have finished this sentence that now reads as “In the following sections, we will describe most of the known proteins (Table 2)” and it is on page 7 lines 306-307 of the revised manuscript.

C. elegans in line 301 should be italicized.  

C. elegans is now italicized in line 356 of the revised manuscript

Round 2

Reviewer 1 Report

No additional comments